# Adaptive Fast Image Encryption Algorithm Based on Three-Dimensional Chaotic System

**DOI:** 10.3390/e25101399

**Published:** 2023-09-29

**Authors:** Yiming Wang, Xiangxin Leng, Chenkai Zhang, Baoxiang Du

**Affiliations:** Electronic Engineering College, Heilongjiang University, Harbin 150080, China; 2211810@s.hlju.edu.cn (Y.W.);

**Keywords:** adaptive control, chaotic system, FPGA, image encryption, security analysis

## Abstract

This paper introduces a novel three-dimensional chaotic system that exhibits diverse dynamic behaviors as parameters vary, including phase trajectory offset behaviors and expansion–contraction phenomena. This model encompasses a broad chaotic range and proves suitable for integration within image encryption. Building upon this chaotic system, the study devised a fast image encryption algorithm with an adaptive mechanism, capable of autonomously determining optimal encryption strategies to enhance algorithm security. In pursuit of heightened encryption speed, an FPGA-based chaotic sequence generator was developed for the image encryption algorithm, leveraging the proposed chaotic system. Furthermore, a more efficient scrambling algorithm was devised. Experimental results underscore the superior performance of this algorithm in terms of both encryption duration and security.

## 1. Introduction

The rapid evolution of mobile information technology has positioned images as pivotal carriers of information in societal communication. However, transmitting images over networks comes with risks. Consequently, addressing security concerns during image transmission has become a significant topic. Given the voluminous nature, high redundancy, and real-time demands of image data, traditional encryption techniques are inadequate for image encryption [1]. Chaotic systems, characterized by complex dynamic behaviors, sensitivity to initial conditions, and long-term unpredictability of their dynamics, offer better pseudorandom properties for signal generation [2]. In [3], high-dimensional conservative chaotic systems were designed using coupled memristors. The chaotic systems exhibited two types of bifurcation enhancement behaviors, In [4], in the reported fractional-order hyperchaotic systems, hidden homomorphism extreme multiple stability and initial offset boosting behavior were found. Ref. [5] designed a multistability and offset-boosting conservative chaotic system exhibiting rich dynamical behaviors. Ref. [6] proposed a four-dimensional chaotic system and validated it with a circuit implementation, proving the physical realizability of the chaotic system. Ref. [7] introduced a conservative dynamical system exhibiting chaos properties and implemented it on an FPGA. Ref. [8] constructed two Hamiltonian conservative chaotic systems and designed a pseudorandom signal generator based on these systems using an FPGA, passing the NIST tests. Moreover, various chaotic systems have exhibited distinct dynamic behaviors such as hyper-chaos [9], multi-stability [10], offset-boosted behavior [11], multi-wing attractors [12], and multi-scroll attractors [13]. These traits underline the inherent advantages of chaotic systems in the domain of encryption [14,15]. Since Matthews first demonstrated the suitability of chaos theory in encryption algorithms in 1989 [16], an increasing number of chaotic image encryption algorithms have been proposed. Ref. [17] combined one-dimensional chaos with DNA coding techniques to design a robust image encryption algorithm that is highly resistant to noise and shear attacks. Ref. [18] presented a bidirectional diffusing DNA encoding encryption algorithm based on a 5-dimensional chaotic system. Combining convolution operations, a highly sensitive encryption algorithm was designed in [19]. Ref. [20] designed a conservative chaotic system with coexisting chaotic-like attractors and developed an image encryption algorithm based on this system, achieving good encryption results. Ref. [21] designed a novel image encryption algorithm with closed-loop diffusion between blocks based on a conservative hyperchaotic system. This scheme can ensure the sensitivity of the encryption system. Experimental results showed this scheme has good encryption effects. Ref. [22] proposed a conservative system based on a triangular wave memristor and applied it in image encryption. This scheme has extremely high sensitivity to initial values and can counter the risk of data reconstruction attacks.

In recent times, an increasing number of encryption algorithms have been proposed, and these algorithms are becoming increasingly complex. Currently, very few encryption methods adapt to the different types of images they are encrypting. The lack of an adaptive mechanism was addressed in [23]. Image encryption algorithms with adaptive mechanisms such as the novel visually meaningful image encryption algorithm based on parallel compressive sensing and adaptive embedding [24] and the meaningful image encryption algorithm based on newly designed coupled map lattice and adaptive embedding [25] have long encryption and decryption processing times, causing them difficulty in meeting real-time encryption needs. Field programmable gate arrays (FPGAs) are reconfigurable integrated circuits that can be programmed to perform specific tasks, making them highly suitable for various complex applications [26]. Moreover, the parallel execution capability of the Verilog language endows FPGAs with advantages such as high speed and parallel processing. Chaotic systems implemented with FPGAs can thus achieve faster speeds. With the continuous advancement of FPGA technologies, an increasing number of chaotic systems have been implemented with FPGAs. Ref. [27] presented a pseudo-random number generator based on a discrete hyper-chaotic system with a bit throughput up to 2.10 Gbps using an FPGA. Ref. [28] proposed a new FPGA-based multi-wing chaotic system. Ref. [29] realized a fractional memristive chaotic system using an FPGA. Ref. [30] developed a 5D memristive exponential hyperchaotic system with an FPGA. Ref. [31] designed a new 3D chaotic system using an FPGA. Additionally, since the entire chaotic system generation process is encapsulated within the encryption unit, it is difficult for attackers to acquire the structure of the chaotic system externally, thus physically ensuring the security of the encryption algorithm [32]. Based on the above analysis, this study combined the fast computing advantages of FPGAs and designed an adaptive fast image encryption algorithm whose main contributions include: (1) proposing a novel three-dimensional chaotic system, which was found to demonstrate diverse phase trajectories, offset behaviors, and expansion–contraction phenomena in response to parameter variations, encompassing a broad chaotic range suitable for integration within image encryption; (2) implementing this chaotic system with an FPGA and designing a chaos sequence generator for image encryption; (3) designing an adaptive fast image encryption algorithm incorporating the chaos sequence generator.

The remainder of this paper is organized as follows: Section 2 introduces the proposed novel three-dimensional chaotic system and presents dynamical analyses, as well as the FPGA implementation of the chaotic system and chaos sequence generator. Section 3 describes the adaptive fast image encryption algorithm designed in this study. Section 4 provides security analyses to evaluate its security and reliability. Section 5 concludes the paper.

## 2. Chaotic System

### 2.1. Proposed 3D Chaotic System

According to the method in [33], we propose a new three-dimensional chaotic system. The mathematical expression of this system is shown in Equation (1)
(1)$$\left\{ \begin{array}{l} \dot{x}=ay+cxz-z\\ \dot{y}=-ex+bz\\ \dot{z}=x-by-cx^{2}+d \end{array}\right.$$

This study elucidates the dynamical behaviors of the conservative chaotic system (1) under periodic and chaotic regimes. The periodic parameter set is specified as *a* = 2, *b* = 4, *c* = 2, *d* = 1, *e* = 2, with the initial condition (1, 1, 1). The chaotic parameter set is given by *a* = 2, *b* = 4, *c* = 2, *d* = 10, *e* = 2, under the identical initial condition. The resulting phase portraits (*x*–*y*, *y*–*z*, *x*–*z* planes) in Figure 1a–c exhibit smooth topologies, indicative of periodic motions, whereas Figure 1d–f manifest irregular traits, signifying chaotic dynamics. The 0–1 test further distinguishes the periodic (Figure 1g) and chaotic (Figure 1h) trajectories. Additionally, the Lyapunov exponents take values (0, 0, 0) and (−0.23, 0, 0.23) for the periodic and chaotic scenarios, respectively, with vanishing sums obeying conservation laws. This substantiates the unique dynamical behaviors of system (1) under the periodic and chaotic regimes.

### 2.2. Different Phase Trajectories Varying with b, c

This study elucidates the dynamical behaviors of system (1) by modulating the control parameters *b* and *c*. The Lyapunov exponent spectrum as a function of *b* within the interval [1,10] is delineated in Figure 2a, For a three-dimensional continuous chaotic system, its Lyapunov exponent is shown in Equation (2) [34]:(2)$$\left\{\begin{array}{c} \mathrm{LE}1=\sigma_{1}=\underset{t\to \infty }{\lim}\frac{1}{t}\ln\frac{\left|\delta x(t)\right|}{\left|\delta x(0)\right|}\\ \mathrm{LE}2=\sigma_{2}=\underset{t\to \infty }{\lim}\frac{1}{t}\ln\frac{\left|\delta y(t)\right|}{\left|\delta y(0)\right|}\\ \mathrm{LE}3=\sigma_{3}=\underset{t\to \infty }{\lim}\frac{1}{t}\ln\frac{\left|\delta z(t)\right|}{\left|\delta z(0)\right|} \end{array}\right.$$
where LE1 ≥ LE2 ≥ LE3 signify the three Lyapunov exponents. Figure 2b illustrates the bifurcation diagram of the local maxima of the variable *x* versus *b*, congruent with the dynamical characteristics embodied in the Lyapunov exponent spectrum. 

To further scrutinize the sensitivity of system (1) to *b*, the *x*–*y* phase portraits for various *b* values are expounded in Figure 3. The topological structures of these conservative flows are distinct from each other. Figure 3 only showcases six types of conservative flow patterns generated by system (1). In fact, myriad other morphologies of conservative flows exist, substantiating that system (1) exhibits abundant dynamical behaviors.

In a similar vein, with regard to the parameter *c*, Figure 4 portrays the Lyapunov exponent spectra and bifurcation diagram encompassing the interval *c* ∈ [1, 10]. The phase portraits in the *x*–*y* plane, delineating the dynamics of system (1) across a range of *c* values, are delineated in Figure 5. Over the course of this progression, system (1) experiences a rhythmic interplay between quasiperiodic and chaotic regimes. Upon a detailed inspection of Figure 5, it becomes discernible that these topological configurations governing the conservative flows exhibit pronounced variations. It is imperative to underscore that Figure 5 exclusively showcases six discrete manifestations of conservative attractors generated by the dynamic interplay of system (1). It is worth noting that the actual array of conservative attractor forms is even more extensive and heterogeneous, thereby accentuating the intricate and multifaceted nature of the system’s dynamical behavior.

### 2.3. Large-Scale Chaos That Varies with d

Through simulations, it has been observed that System (1) possesses an extensive region of chaotic behavior. Let *d* be the control variable; the Lyapunov exponent spectrum and bifurcation diagram depicting variations with respect to *d* are presented in the Figure 6. Notably, the entire range of Lyapunov exponents (LEs) within this interval is positive, indicating chaotic motion. This correspondence with positive LEs is also reflected in the irregular points of the bifurcation diagram. It is important to highlight that the figure displays only a partial interval (*d* ranging from 4 to 20). In practice, when d exceeds 20, System (1) consistently exhibits chaotic behavior.

### 2.4. Offset Behavior That Varies with e

With the manipulation of the control parameter *e*, the dynamical trajectories of System (1) demonstrate distinct displacement patterns along the x and z axes. By adopting an incremental interval of 1, a selection of five parameter sets is curated, initiating from a value of 3. Figure 7 portrays the *x*–*z* phase trajectory plots corresponding to *e* values of 3, 4, 5, 6, and 7. Evidently, as the control parameter e escalates, the phase trajectories depicted in Figure 7 undergo varying magnitudes of deviation in both the *x* and *z* dimensions, cohesively materializing as motion along the diagonal axis.

### 2.5. Contour Shrinks as x Expands and Contracts

For system (1) under periodic regimes with initial conditions (*x*_0_, 1, 1), seven *x*_0_ values of 0, ±1, ±2, ±3 are prescribed. The consequent phase trajectory evolutions are elucidated in Figure 8a,b. Astute examination with 0 as the critical point (purple phase portrait) unveils the pattern upon varying *x*_0_. When *x*_0_ is negative, the global contour of the phase trajectories contracts with increasing *x*_0_. Likewise, when *x*_0_ is positive, the overall contour shrinks as *x*_0_ increases. That is, the phase portraits expand with escalating absolute value of *x*_0_.

### 2.6. Complexity

Complexity serves as a robust indicator for assessing the degree of randomness within sequences. Currently, a plethora of algorithms are employed to gauge the complexity of testing systems, including spectral entropy (SE), wavelet entropy (WE), Kolmogorov entropy, and the C0 complexity algorithm, among others [35,36,37,38,39]. In order to assess the complexity of the present system, this study employs both SE and the C0 complexity algorithm to determine their suitability for information security applications. The SE algorithm [40] utilizes the energy distribution in the Fourier transform domain. Based on the Shannon entropy algorithm, the spectral entropy value is obtained. The algorithm is shown in Equation (3)
(3)$$SE\left(N\right)=\frac{-\sum_{k=0}^{\frac{N}{2-1}}P_{k}\ln P_{k}}{\ln\left(\frac{N}{2}\right)}$$
where $P_{k}=\frac{{\left|X(k)\right|}^{2}}{\sum_{k=0}^{\frac{N}{2}-1}{\left|X(k)\right|}^{2}}$, X(k) is Perform discrete Fourier transform on Equation (4):(4)$$x\left(n\right)=x\left(n\right)-\frac{1}{N}\sum_{n=0}^{N-1}x(n)$$

The specific calculation steps of the C0 complexity algorithm are shown in Equation (5) [41]:(5)$$C0(r,N)=\sum_{n=0}^{N-1}{\left|x\left(n\right)-\tilde{x}(n)\right|}^{2}/\sum_{n=0}^{N-1}{\left|x(n)\right|}^{2}$$
with higher values indicating greater system complexity. Figure 9 is constructed with *x*_0_ and *y*_0_ as control variables and parameters *a* = 2, *b* = 4, *c* = 2, *d* = 10, and *e* = 2. The initial condition is set as (*x*_0_, *y*_0_, 0.2), where *x*_0_ ∈ [−3, 3] and *y*_0_ ∈ [−3, 3]. The extensive regions with elevated numerical values depicted in the figure signify the system’s profound complexity, rendering it suitable for employment in image encryption endeavors.

### 2.7. NIST Test

To test the randomness of the chaotic sequences generated by the chaotic system (1) in the encryption algorithm, the key is injected into system (1) for iteration M × N (M × N is related to the size of the encrypted image). Afterwards, randomness tests are performed on the generated sequences. In this section, the SP800-22 Rev1a standard is adopted for testing. The resulting *p*-values from these tests are presented in Table 1. As per the standard, sequences are deemed to possess randomness when their *p*-values exceed 0.01. If the tested sequence successfully passes all tests, its robust randomness is substantiated [42,43].

### 2.8. Key Sensitivity

Figure 10 presents the key sensitivity test results for system (1). By introducing minute perturbations of 10^−15^ to the initial value x_0_ and comparing the resultant sequences after 50 iterations, it is evident that system (1) exhibits high sensitivity to initial conditions, rendering it suitable for image encryption applications.

### 2.9. FPGA Chaotic Sequence Generator Design

In this study, the proposed three-dimensional chaotic system is implemented on an FPGA as a chaotic sequence generator. To validate the correctness of the FPGA implementation, the generated chaotic sequences are output to an oscilloscope for analysis. The Cyclone IV E EP4CE10F17C8 is utilized as the main control chip to drive the dual-channel digital-to-analog converter DAC8552 for analog signal output. After verification, the chaotic sequences are transmitted to the encryption system via a serial communication circuit, with the computation controlled by push buttons. The hardware design scheme is depicted in Figure 11.

#### 2.9.1. FPGA Chaotic System Validation

The Euler algorithm, as a means of solving differential equations [44], can be utilized to discretize chaotic systems [45]. Ref. [46] deployed a chaotic system on an FPGA using the Euler algorithm. However, due to the high sensitivity of this system to initial values, higher computational precision is required. The use of the Euler algorithm may lead to degradation of chaos. An improved Euler algorithm can effectively improve the computational accuracy [47,48]. In this paper, the improved Euler algorithm is adopted to discretize the system (1). The circuit is designed based on the discretized chaotic system. The discretized expressions are shown in (6) to (10):(6)$$\left\{\begin{array}{c} {\;H}_{1}=x_{0}\;\\ {\;H}_{2}=y_{0}\\ \;H_{3}=z_{0}\; \end{array}\right.$$
(7)$$\left\{\begin{array}{c} J_{1}=a\times {\;H}_{2}+c{\times H}_{1}\times H_{3}-H_{3}\\ J_{2}=-e\times H_{1}+b\times H_{3}\\ J_{3}=-b{\times H}_{2}-c{\times H}_{1}\times H_{1}+d+H_{1} \end{array}\right.$$
(8)$$\left\{\begin{array}{c} K_{1}=H_{1}+0.005\times J_{1}\\ K_{2}=H_{2}+0.005\times J_{2}\\ K_{3}=H_{3}+0.005\times J_{3} \end{array}\right.$$
(9)$$\left\{\begin{array}{c} L_{1}=a\times K_{2}+c\times K_{1}\times K_{3}-K_{3}\\ L_{2}=-e\times K_{1}+b\times K_{3}\\ L_{3}=-b\times K_{2}-c\times K_{1}\times K_{1}+d+K_{1} \end{array}\right.$$
(10)$$\left\{\begin{array}{c} Y_{1}=H_{1}+0.01\times L_{1}\\ Y_{2}=H_{2}+0.01\times L_{2}\\ Y_{3}=H_{3}+0.01\times L_{3} \end{array}\right.$$

In Equation (2), *x*_0_, *y*_0_, *z*_0_ denote the required initial values for the system. 

The chaotic system is implemented in Verilog HDL; the actual connections are shown in Figure 12, with the generated logic structure wiring diagram shown in Figure 13. U1 is a floating-point adder module, U2 is a floating-point multiplier module, U0 is an iterative distribution module to realize Equations (6)–(10), U3 is a floating-point to fixed-point conversion module, U6 performs timing synchronization, and U7 drives the DAC8552.

The phase trajectories obtained from MATLAB simulation are shown in Figure 1a–f, Figure 3a–c and Figure 5b,d,e. With the same parameters, the phase portraits realized on the FPGA are depicted in Figure 14. By comparing with the simulated phase trajectories, it is validated that the FPGA can successfully implement system (1), demonstrating the accuracy of the FPGA chaotic system implementation.

#### 2.9.2. Pseudo-Random Sequence Generator Design

This subsection extends the design presented in Section 2.9.1 by incorporating a chaos sequence processing module (called “prng”) and a serial transmission module (called “uart_send”) to construct a chaos sequence generator. The chaos sequence processing module disassembles the three-dimensional chaotic sequence into 8-bit data segments, in accordance with the protocols delineated in Table 2 and Figure 15. This segmentation is undertaken to facilitate handling of integer values ranging from 0 to 255. Subsequently, the disintegrated data are cyclically emitted through a state machine-based mechanism.

The serial transmission module implements serial communication to transmit the processed random numbers. As shown in Figure 16, you can see the data received on the PC side through the serial debugging tool.

To verify the randomness of the data generated by the chaotic sequence generator, the SP800-22 Rev1a test suite was utilized to test the sequences produced by the chaotic sequence generator. SP800-22 Rev1a recommends using test sequence lengths between 10^3^ and 10^7^. In this section, a test sequence length of 10^6^ was chosen. The test results are shown in Figure 17, demonstrating that the sequences generated by the chaotic sequence generator are random and suitable for use in image encryption. The generated logic structure wiring diagram is shown in Figure 18

## 3. Encryption Algorithm

This section elucidates the encryption algorithm adopted in this study. In the pre-encryption stage, the raw image is pre-processed into a 64 × 64 grayscale image to meet pre-encryption requirements. And in the pre-encryption stage, exhaustive exploration of all possible block schemes is conducted. The optimal encryption strategy for the current state is determined via pertinent evaluation functions, with the optimal solution being fed back into the encryption system. Simultaneously, the optimal solution is embedded into the least significant bit of the encrypted image. Lastly, a fast scrambling algorithm is designed to further encrypt the image. The overall encryption scheme is illustrated in Figure 19.

### 3.1. Chunk Encryption Algorithm

This paper proposes a chunk encryption algorithm designed to further reduce the computational time of the encryption algorithm. The specific procedure of the chunk algorithm is outlined as follows:

Step 1: The original image A is read and its dimensions are obtained, where M is the image length and N is the image width.

Step 2: Extract the RGB components of the color image.

Step 3: A pseudo-random sequence generator is utilized to produce the random matrix B with the same dimensions as the image.

Step 4: Inject the initial parameters of the chaotic system as the key simultaneously into the pseudorandom sequence generator and system (1) to generate the pseudorandom matrix B and pseudorandom sequences *x*, *y*, *z* required for encryption.

Step 5: Perform chunking on image A and pseudorandom matrix B, where the chunking process is shown in Figure 20. The parameter F controls the chunk size.

Step 6: Substitute the chunked image A and chunked pseudorandom sequence B into Equation (11) to encrypt the pixel values of image A.
(11)$$D_{i}=\left\{\begin{array}{c} (A_{i}+B_{i}+A_{M})\mathrm{mod}256,\;\;i=1\\ \left(A_{i}+B_{i}+D_{i-1}\right)\mathrm{mod}256,\;\;i\in \left[2,M\right] \end{array}\right.$$

In the equation, $A_{i}$ represents the image data matrix of size F × M after partitioning, $B_{i}$ represents the pseudorandom matrix of size F × M after partitioning, and $D_{i}$ represents the encrypted data. 

When encrypting the first chunk of data, the first chunk of A, the last chunk of A, and the first chunk of B need to be added together modulo and then encrypted to the first chunk of D. When encrypting other chunks of data, the current chunk of A, the current chunk of B, and the previous chunk of D are used for encryption. Figure 21 helps better understand the algorithm equation above. Figure 22 demonstrates the comparison of encryption time for point-by-point encryption and different chunking strategies, It can be found. During encryption, matrix operations have faster encryption speeds compared to point-by-point operations.

Step 7: Repeat Steps 4 through 6 for the encryption of the RGB components.

Impact of Different Chunking Strategies on Encryption Performance

During the design phase, we observed that different block strategies result in varying encryption outcomes for different images. This study conducts distinct evaluations for blocks of sizes 1, 2, 4, 8, 16, 32, and 64 rows, respectively. The evaluation encompasses encrypted information entropy, correlation, resistance against differential attacks, encryption time, and a comparison of structural similarity before and after encryption. In order to vividly portray the encryption performance under distinct block strategies, this paper performs data normalization across all datasets. The encryption performance of various images under different block strategies is depicted in Figure 23.

### 3.2. Adaptive Chunking Design

To determine the optimal encryption strategy, an adaptive block-based encryption process is designed to select different block schemes based on the image for improved encryption performance. The specific steps are:

Step 1: The image is resized to a 64 × 64 grayscale image. 

Step 2: Pre-encryption is performed using the block diffusion algorithm in Section 3.1.

Step 3: All block schemes are iterated, and the entropy, correlation, encryption time, and structural similarity before and after encryption are computed for each scheme.

Step 4: An evaluation function assesses the overall performance of each scheme and derives the optimal solution.

Step 5: The optimal solution is embedded in the encrypted image matrix for decryption.

#### 3.2.1. Data Concealment Design

In the phase of data concealment design, the optimal solution is embedded within a set of data by utilizing the least significant bits. Given the maximum block strategy of 64 in the design, 7 bits of data are required to conceal the optimal solution. A matrix W of size 7 × M is acquired using a chaotic sequence generator, where M represents the length of the image. Both matrix W and the optimal solution are converted into binary form. Each column within matrix W corresponds to a group, and the optimal solution is written into the least significant bit of each group, thereby achieving data concealment. The schematic is shown in Figure 24.

Specifically, the 1st to 7th values of the chaotic sequence *z* are selected and processed using Equation (12) to yield FZ.
(12)$$FZ=\left\lceil \left(Z\times 2^{16}\right)modM\right\rceil$$

Each row of the steganographic matrix W is inserted into the FZ rows of the image encryption matrix. Through the utilization of a scrambling algorithm, this matrix can be concealed within the image data encryption matrix.

#### 3.2.2. Evaluation Function Design

The evaluation function incorporates four key assessment metrics: information entropy (E1), correlation (C1), time (T1), and structural similarity (S1). In terms of encryption effectiveness, we strive for information entropy (E1) to approach 8 for superior performance, correlation (C1) to approximate zero for better results, minimal time consumption indicated by (T1), and reduced structural similarity (S1) for enhanced performance. Guided by these evaluation criteria, a decision matrix, as illustrated in Table 3, has been formulated.

To mitigate the influence of dimensions, it is essential to subject the four variables mentioned above to standardized processing. Utilizing normalization, each of the indicators is mapped onto the interval [0, 1], Furthermore, Equation (13) is employed to compute the Euclidean distance between the evaluation metrics produced by each block strategy and the ideal as well as negative ideal values.
(13)$$\left\{\begin{array}{c} D_{+}=\sqrt{{\left(E_{max}-E1\right)}^{2}+{\left(C_{max}-C1\right)}^{2}+{\left(T_{max}-T1\right)}^{2}+{\left(S_{max}-S1\right)}^{2}}\\ D_{-}=\sqrt{{\left(E_{min}-E1\right)}^{2}+{\left(C_{min}-C1\right)}^{2}+{\left(T_{min}-T1\right)}^{2}+{\left(S_{mon}-S1\right)}^{2}} \end{array}\right.$$

The relative proximity gradient (R) for each block strategy is subsequently computed using Equation (14).
(14)$$R=\frac{D_{-}}{D_{-}+D_{+}}$$

In conclusion, the R values derived from all block strategies are arranged in ascending order, with the block strategy yielding the highest R value being selected as the optimal solution.

### 3.3. Scrambled Encryption Algorithm

In the context of the scrambling encryption process, we have devised a rapid algorithm inspired by Rubik’s Cube shifting. The image matrix achieves rapid scrambling through a combination of row and column shifts, where the shift stride is controlled by a chaotic system and the shifting direction is determined by the pixel points within the image. Hereinafter, we will expound on the devised scrambling encryption algorithm, incorporating Figure 25. 

Step 1: Read the block encryption matrix D and ascertain the dimensions of the image to be encrypted. Let M represent the image length, and N the image width.

Step 2: Extract the RGB components of the encryption matrix D.

Step 3: Employ the initial parameters of the chaotic system as the key input into Equation (1). Discard the first 500 points to eliminate transient effects of the chaotic system, yielding the chaotic sequences x1 and x2. 

Step 4: Obtain the key parameter n and extract n to M + n position from the x sequence to control row shifts. Apply Equation (15) to process the chaotic sequence x, engendering a pseudo-random sequence Z1 of length M, varying within the range of 0 to M. Here, M denotes the image width.
(15)$$\left\{\begin{array}{c} X1=x_{i}\;\;\;i\in (n,M+n)\\ Z1=\left\lceil (X1\times 2^{16})modM\right\rceil \end{array}\right.$$

Step 5: Extract M + n + 1 to 2M + n position from the x sequence and substitute them into Equation (16) to beget a sequence L1 of length M−1.
(16)$$\left\{\begin{array}{c} X2=x_{i}\;\;\;i\in (M+n+1,\;2M+n)\\ L1=\left\lceil \left(X2\times 2^{16}\right)modM\right\rceil \end{array}\right.$$

Step 6: Perform cyclic shift processing on the first row of image matrix D using Equation (17).
(17)$${PB}_{i}=\left(D>>>\left(N-{Z1}_{i}\right)|\left(D_{i}<<<{Z1}_{i}\right)\right),i=1$$

Step 7: The L1 value is used to select a pixel from the previous row. The numeric value of this pixel controls the shift direction, with odd numbers corresponding to cyclic right shifts, and even numbers to cyclic left shifts, processed by Equation (18).
(18)$$\left\{\begin{array}{c} {PB}_{i}=\left(D_{i}>>>\left(N-{Z1}_{i}\right)|\left(D_{i}<<<{Z1}_{i}\right)\right),\mathrm{if}\;D\left(i-1,{L1}_{i}\right)=2n,i\in \left[2,M\right]\\ {PB}_{i}=\left(D_{i}<<<\left(N-{Z1}_{i}\right)|\left(D_{i}>>>{Z1}_{i}\right)\right),if\;D\left(i-1,{L1}_{i}\right)=2n+1,i\in \left[2,M\right] \end{array}\right.$$

Step 8: Similarly, bits n to N + n of the *y* sequence are extracted to control column shifts. Substituting into Equation (19) gives the shift strides Z2 required for column shifts, where N denotes image width.
(19)$$\left\{\begin{array}{c} X3=y_{i}\;i\in (n,N+n)\\ Z2=\left\lceil (X3\times 2^{16})modN\right\rceil \end{array}\right.$$

Step 9: Bits N + n + 1 to 2N + n of y are taken and applied in Equation (20) to generate the parameter L2 controlling column shift directions.
(20)$$\left\{\begin{array}{c} X4=y_{i}\;i\in (N+n+1,2N+n)\\ L2=\left\lceil (X4\times 2^{16})modN\right\rceil \end{array}\right.$$

Step 10: Cyclic column shifts are performed on each column of matrix PB using Equation (21) to obtain the column-shifted matrix AA.
(21)$$\left\{\begin{array}{c} {AA}_{j}=\left({PB}_{j}>>>\left(M-{Z2}_{j}\right)|\left({PB}_{j}<<<{Z2}_{j}\right)\right),j=1\\ {AA}_{j}=\left({PB}_{j}>>>\left(M-{Z2}_{j}\right)|\left({PB}_{j}<<<{Z2}_{j}\right)\right),\mathrm{if}\;PB\left({L2}_{j},j-1\right)=2n,j\in [2,N]\\ {AA}_{j}=\left({PB}_{j}<<<\left(M-{Z2}_{j}\right)|\left({PB}_{j}>>>{Z2}_{j}\right)\right),if\;PB\left({L2}_{j},j-1\right)=2n+1,j\in [2,N] \end{array}\right.$$

Step 11: Repeat Steps 4–10 to scramble the RGB components. 

Figure 26 shows a comparison with several conventional algorithms, which indicates that the proposed algorithm is the least time-consuming and is suitable to be applied to achieve fast image encryption.

### 3.4. Decryption Algorithm

Decryption is the inverse process of encryption, and the overall flow of decryption is shown in Figure 27.

First, the encrypted image must undergo scrambling decryption. The encryption matrix AA is column-decrypted using Equation (22) to obtain matrix PBK:(22)$$\left\{\begin{array}{c} {PBK}_{j}=\left({AA}_{j}<<<\left(M-{Z2}_{j}\right)|\left({AA}_{j}<<<{Z2}_{j}\right)\right),j=1\\ {PBK}_{j}=\left({AA}_{j}<<<\left(M-{Z2}_{j}\right)|\left({AA}_{j}>>>{Z2}_{j}\right)\right),if\;P\left({L2}_{j},j-1\right)=2n,j\in [2,N]\\ {PBK}_{j}=\left({AA}_{j}>>>\left(M-{Z2}_{j}\right)|\left({AA}_{j}<<<{Z2}_{j}\right)\right),if\;P\left({L2}_{j},j-1\right)=2n+1,j\in [2,N] \end{array}\right.$$

Then, Equation (23) is utilized to acquire the row decryption matrix DZ:(23)$$\left\{\begin{array}{c} {DZ}_{i}=\left({PBK}_{i}<<<\left(N-{Z1}_{i}\right)|\left({PBK}_{i}<<<{Z1}_{i}\right)\right),i=1\\ {DZ}_{i}=\left({PBK}_{i}<<<\left(N-{Z1}_{i}\right)|\left({PBK}_{i}>>>{Z1}_{i}\right)\right),if\;P\left({i-1,L1}_{i}\right)=2n,i\in [2,M]\\ {DZ}_{i}=\left({PBK}_{i}>>>\left(N-{Z1}_{i}\right)|\left({PBK}_{i}<<<{Z1}_{i}\right)\right),if\;P\left({i-1,L1}_{i}\right)=2n+1,i\in [2,M] \end{array}\right.$$

After scrambling decryption, matrix DZ is utilized to extract the hidden matrix W via Equation (8). The optimal solution F is restored based on the rules in Section 3.2.1 for subsequent decryption. Note that the restored data may have missing or altered values. The mode value from a data group is chosen as the optimal solution F for the next step of decryption.

Finally, further decryption of the encryption matrix DZ is conducted using Equation (24) to acquire the decrypted image I:(24)$$I_{i}=\left\{\begin{array}{c} \left({DZ}_{i}-{DZ}_{i-1}-B_{i}\right)mod256,\;\;if\;i\in [2,M]\\ \left({DZ}_{i}-I_{M}-B_{i}\right)mod256,\;\;\;\;\;\;\;\;\;\;if\;i=1 \end{array}\right.$$

## 4. Security Analysis

### 4.1. The Results of Encryption and Decryption

To assess the encryption performance of this algorithm, we conducted tests using images of varying sizes, as shown in Table 4. The experimental testing was performed on the Matlab R2023a platform. The results of the experimental tests are illustrated in Figure 28.

### 4.2. Histogram Analysis

Figure 29 depicts the three-dimensional histograms of different images before and after encryption. It is evident that the pixel distribution becomes uniform after encryption, validating the algorithm’s robustness against statistical analysis attacks.

### 4.3. $\chi^{2}$
*Test*

Furthermore, we can assess the histogram distribution based on testing. According to Equation (25), we computed the image, and when the calculated result is less than 293.2483, it indicates that the distribution of the encrypted image on the histogram is relatively uniform.
(25)$$\chi^{2}=\sum_{i=0}^{M*N}\frac{{\left(K_{i}-S_{i}\right)}^{2}}{S_{i}}$$

The encrypted images were tested, and the results are shown in Table 5.

### 4.4. Information Entropy

The information entropy of the image can be calculated using Equation (26):(26)$$H=-\sum_{i=0}^{N-1}P(S_{i})\log_{2}P(S_{i})$$

The closer the encryption result is to 8, the higher the security of the encryption algorithm. Table 6 provides the entropy of the test images, and the entropy comparison with other algorithms is presented in Table 7. Figure 30 offers a more visual depiction of the comparative results.

### 4.5. Adjacent Pixel Correlation 

Equations (27)–(30) are utilized to calculate the correlation coefficients between adjacent pixels. The closer the correlation coefficients of the ciphertext are to 0, the higher the level of security.
(27)$$\rho \left(x,y\right)=\frac{cov(x,y)}{\sqrt{D(x)}\sqrt{D(y)}}$$
where
(28)$$Cov\left(x,y\right)=\frac{1}{N}\sum_{i=1}^{N}\left[x_{i}-E\left(x\right)\right][y_{i}-E(y)]$$
(29)$$D\left(x\right)=\frac{1}{N}\sum_{i=1}^{N}{\left[x_{i}-E(x)\right]}^{2}$$
(30)$$E(x)=\frac{1}{N}\overset{N}{\underset{i=1}{\sum }}x_{i}$$

By evaluating the correlations before and after encrypting the Lena image as depicted in Figure 31, it becomes visually evident that the correlations in various directions are significantly reduced after encryption. Table 8 provides a comparison of correlations with other encryption algorithms.

### 4.6. Differential Attack Analysis

In this study, the encrypted images underwent differential attack testing. The computation formulas for differential attack resistance are respectively shown in Equations (31)–(33).
(31)$$NPCR=\frac{\overset{M}{\underset{i=1}{\sum }}\overset{N}{\underset{j=1}{\sum }}D(i,j)}{M\times N}\times 100\%$$
(32)$$\mathrm{UACI}=\frac{\sum_{i=1}^{M}\sum_{j=1}^{N}\left|P_{1}(i,j)-P_{2}(i,j)\right|}{255\times M\times N}\times 100\%$$
where
(33)$$D(i,j)=\left\{ \begin{array}{l} 0P_{1}(i,j)=P_{2}(i,j)\\ 1P_{1}(i,j)\neq P_{2}(i,j) \end{array}\right.$$

The theoretically ideal values of UACI and NPCR, computed according to the aforementioned equations, are 33.4635% and 99.6094% respectively. For this study, three test images of the same size were chosen. Among these images, 20 randomly selected pixel values were subjected to testing. The calculated NPCR and UACI values are presented in Figure 32. The average NPCR and UACI values for 20 points in the Lena image are compared with those reported in other studies in Table 9.

### 4.7. Noise Attack

In this paper, we simulated the noise attack by adding different densities of noise to the ciphertext image, and Figure 33 shows the decryption effect after adding different densities of noise.

Here, we introduce the Peak Signal-to-Noise Ratio (PSNR) and Structural SIMilarity (SSIM) metrics to evaluate the effectiveness of encryption and decryption on images subjected to noise. A higher PSNR value indicates better image quality, while an SSIM value closer to 1 indicates superior image quality, as displayed in Table 10. It can be observed that the algorithm exhibits a certain degree of noise resistance.

### 4.8. Occlusion Attack 

During transmission, images may also be subjected to malicious occlusion. This study verifies the algorithm’s resilience against occlusion by decrypting ciphertext images that have been occluded to different extents. Figure 34 illustrates the decryption outcomes for images occluded to varying degrees. Table 11 provides evaluation values using SSIM and PSNR metrics for the decrypted images after occlusion. It can be observed that the algorithm possesses a certain degree of occlusion resistance.

### 4.9. Opting for a Plaintext Attack

The encrypted results of entirely black and entirely white plaintext images are depicted in Figure 35. Table 12 presents the calculated results for the ciphertext images’ entropy, chi-square test, and correlation coefficients between adjacent pixels. From the outcomes, it is evident that the encryption algorithm in this study is effective in encrypting both entirely black and entirely white plaintext images. Additionally, within the encryption process, this study designed algorithms related to the plaintext to counteract the occurrence of equivalent keys [55], thereby thwarting chosen-plaintext attacks.

### 4.10. Encryption Time

To assess the encryption speed of the algorithm proposed in this study, experimental testing was conducted using Matlab R2023a on a system with the following specifications: CPU: AMD Ryzen 7 5800H, RAM: 16.0 GB. Images of different sizes and types were used to test the encryption time, and the results are presented in Table 13. A comparison with results from other studies is shown in Table 14. Figure 36 provides a more visual representation of the comparative outcomes.

## 5. Conclusions

This paper introduces a three-dimensional chaotic system that demonstrates sensitivity to parameter variations, exhibiting diverse phase trajectories, offset behaviors, and expansion–contraction phenomena. The system encompasses a wide range of chaotic regions. The generated sequences pass the NIST tests, rendering them suitable for image encryption. Moreover, the proposed chaotic system is implemented using FPGA, and a chaotic sequence generator is designed for image encryption. In terms of encryption algorithm design, this paper presents a fast image encryption algorithm with an adaptive mechanism. Through preprocessing before encryption, the optimal encryption strategy is selected, and a rapid scrambling algorithm is devised. By collaborating with the chaotic sequence generator, rapid image encryption can be achieved. Furthermore, security analysis of the encrypted images reveals the algorithm’s capability to effectively counter differential attacks, cropping attacks, noise attacks, chosen-plaintext attacks, and statistical analysis attacks. The encrypted images display an information entropy close to 8, and the correlation coefficients approach 0, confirming the algorithm’s security. Through comparison with different algorithms, the encryption algorithm proposed in this paper demonstrates superior processing speed, enabling encryption of large amounts of data within a short time. It can be applied for video, real-time images, and various other encryption scenarios.

## Figures and Tables

**Figure 1 entropy-25-01399-f001:**
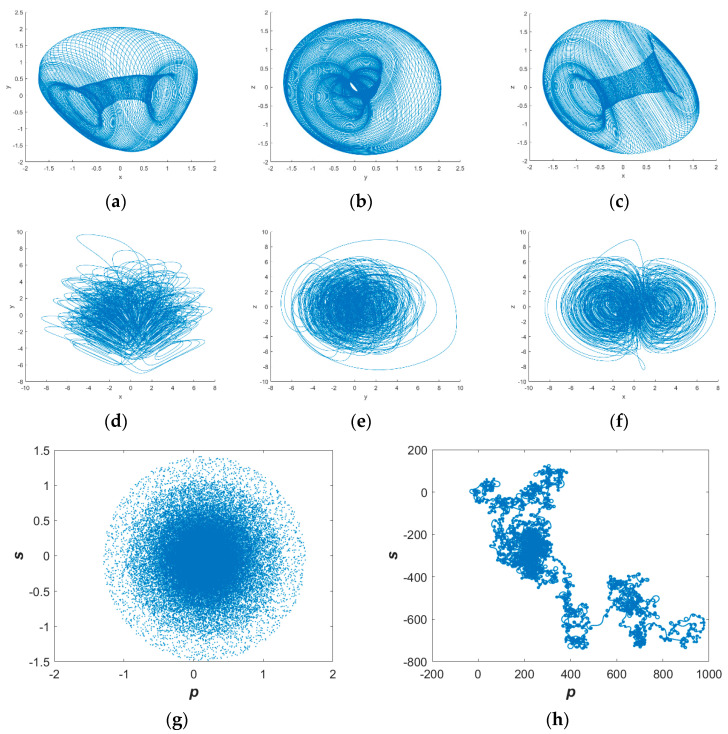
Basic phase diagram of the system: (**a**) *x–y* Plane (period); (**b**) *y*–*z* Plane (period); (**c**) *x*–*z* Plane (period); (**d**) *x*–*y* Plane (Chaos); (**e**) *y*–*z* Plane (Chaos); (**f**) *x*–*z* Plane (Chaos); (**g**) 0–1 test (periodic); (**h**) 0–1 test (chaotic).

**Figure 2 entropy-25-01399-f002:**
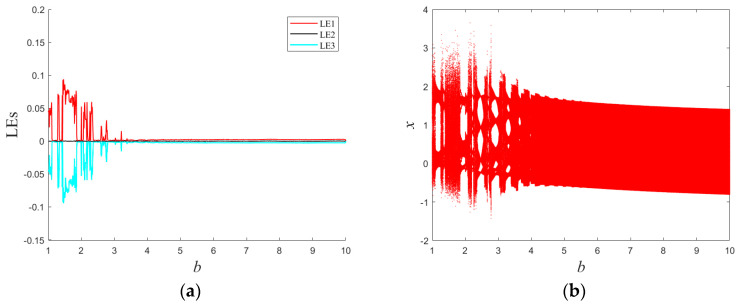
Lyapunov exponential spectrum and bifurcation diagram with *b*: (**a**) Lyapunov exponential spectra; (**b**) bifurcation diagram.

**Figure 3 entropy-25-01399-f003:**
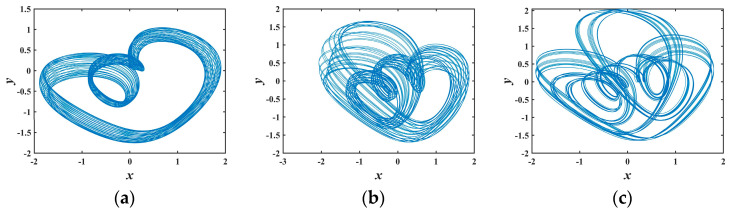
Various phase diagrams varying with *b*: (**a**) *b* = 1.2; (**b**) *b* = 1.9; (**c**) *b* = 2.2; (**d**) *b* = 2.4; (**e**) *b* = 2.8; (**f**) *b* = 4.6.

**Figure 4 entropy-25-01399-f004:**
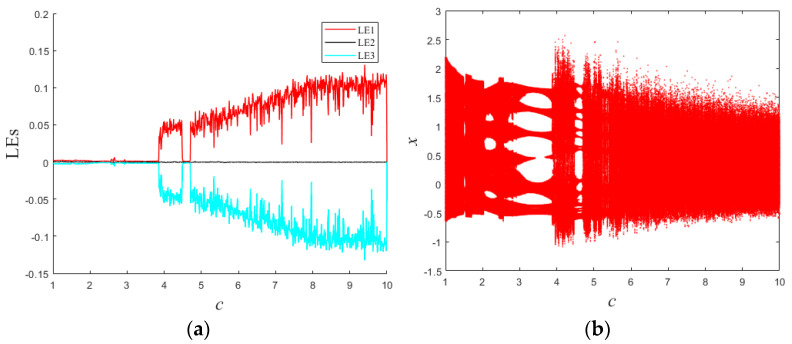
Lyapunov exponential spectrum and bifurcation diagram with *c*: (**a**) Lyapunov exponential spectrum; (**b**) bifurcation diagram.

**Figure 5 entropy-25-01399-f005:**
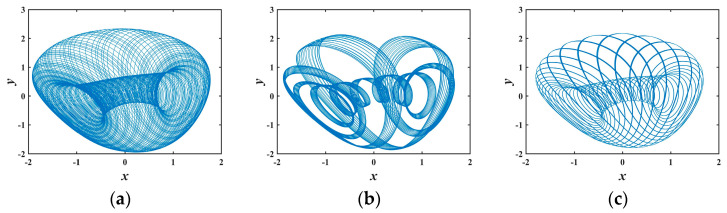
Various phase diagrams varying with *c*: (**a**) *c* = 1.4; (**b**) *c* = 1.5; (**c**) *c* = 1.7; (**d**) *c* = 3.3; (**e**) *c* = 4; (**f**) *c* = 8.5.

**Figure 6 entropy-25-01399-f006:**
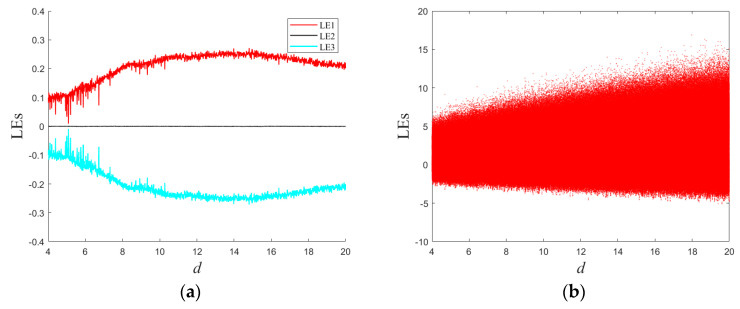
Lyapunov exponential spectra and bifurcation diagram with *d*: (**a**) Lyapunov exponential spectra; (**b**) bifurcation diagram.

**Figure 7 entropy-25-01399-f007:**
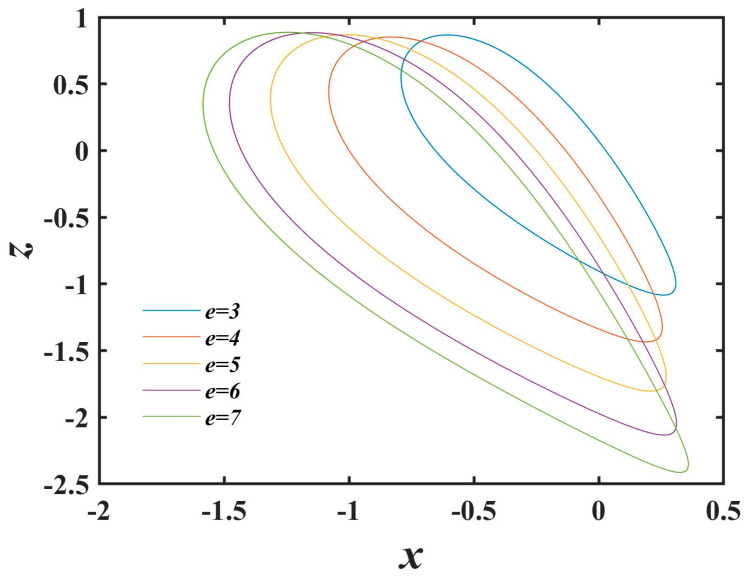
Phase trajectories varying with *e.*

**Figure 8 entropy-25-01399-f008:**
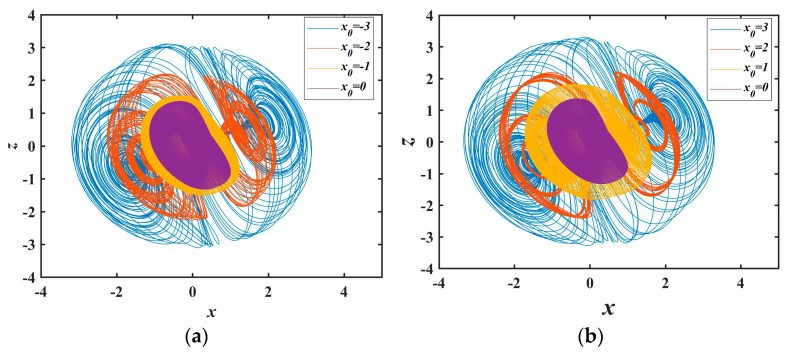
Expansion and contraction behavior with *x*_0_: (**a**) contraction behavior; (**b**) expansion behavior.

**Figure 9 entropy-25-01399-f009:**
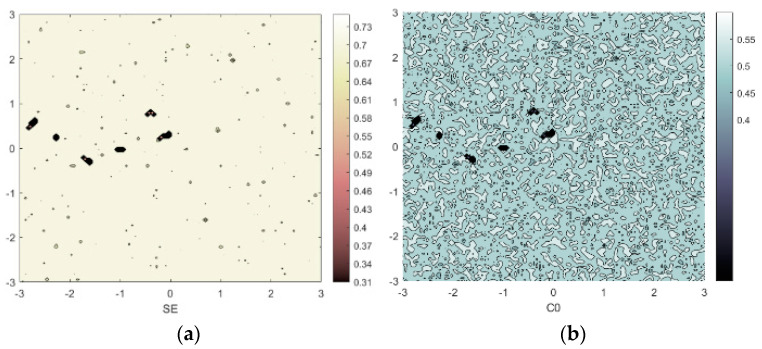
SE and C0 complexity under condition *a* = 2; *b* = 4; *c* = 2; *d* = 10; *e* = 2; (**a**) SE (*x*_0_ ∈ [−3, 3], *y*_0_ ∈ [−3, 3]); (**b**) C0 (*x*_0_ ∈ [−3, 3], *y*_0_ ∈ [−3, 3]).

**Figure 10 entropy-25-01399-f010:**
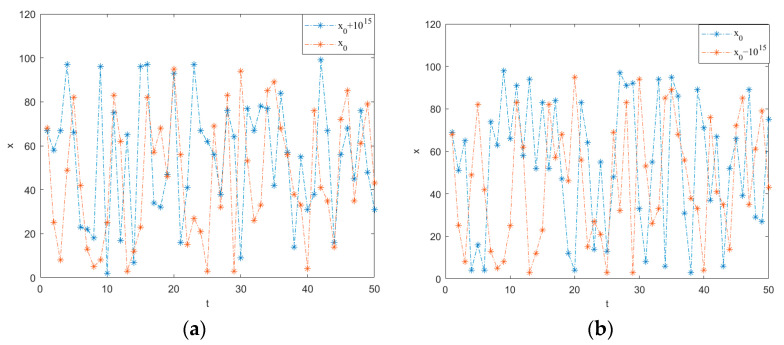
Sensitivity test results for system (1). (**a**) x_0_ + 10^−15^; (**b**) x_0_ − 10^−15^.

**Figure 11 entropy-25-01399-f011:**
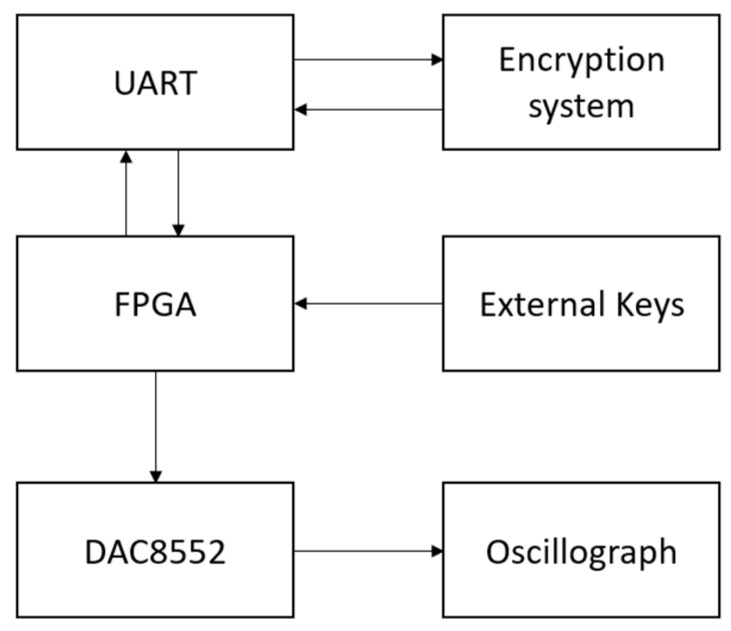
Hardware platform design diagram.

**Figure 12 entropy-25-01399-f012:**
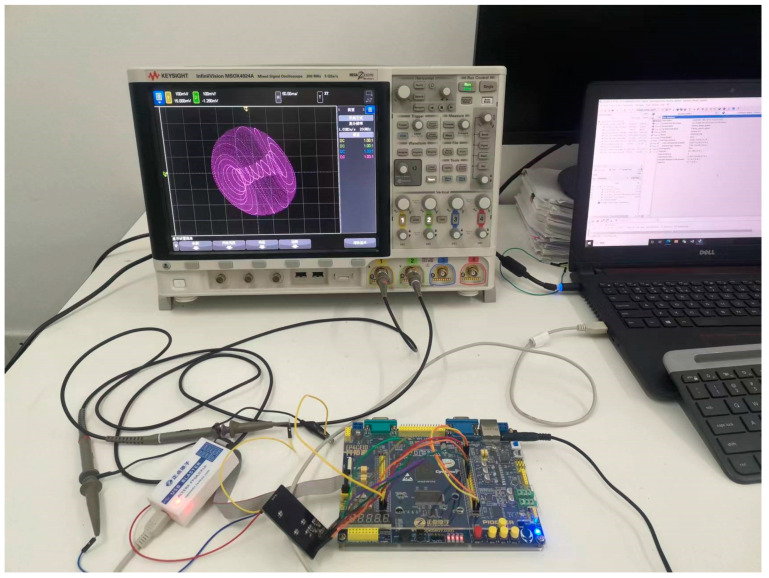
Actual connection setup.

**Figure 13 entropy-25-01399-f013:**
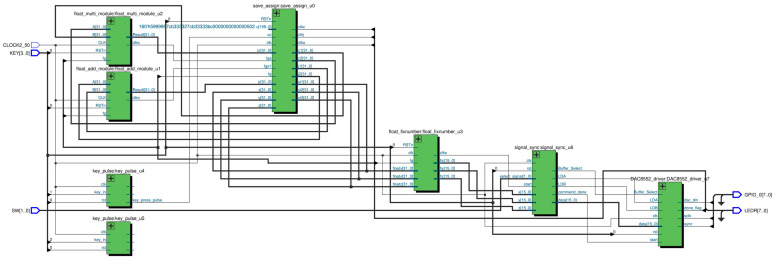
Wiring diagram of the logic structure of the chaotic system.

**Figure 14 entropy-25-01399-f014:**
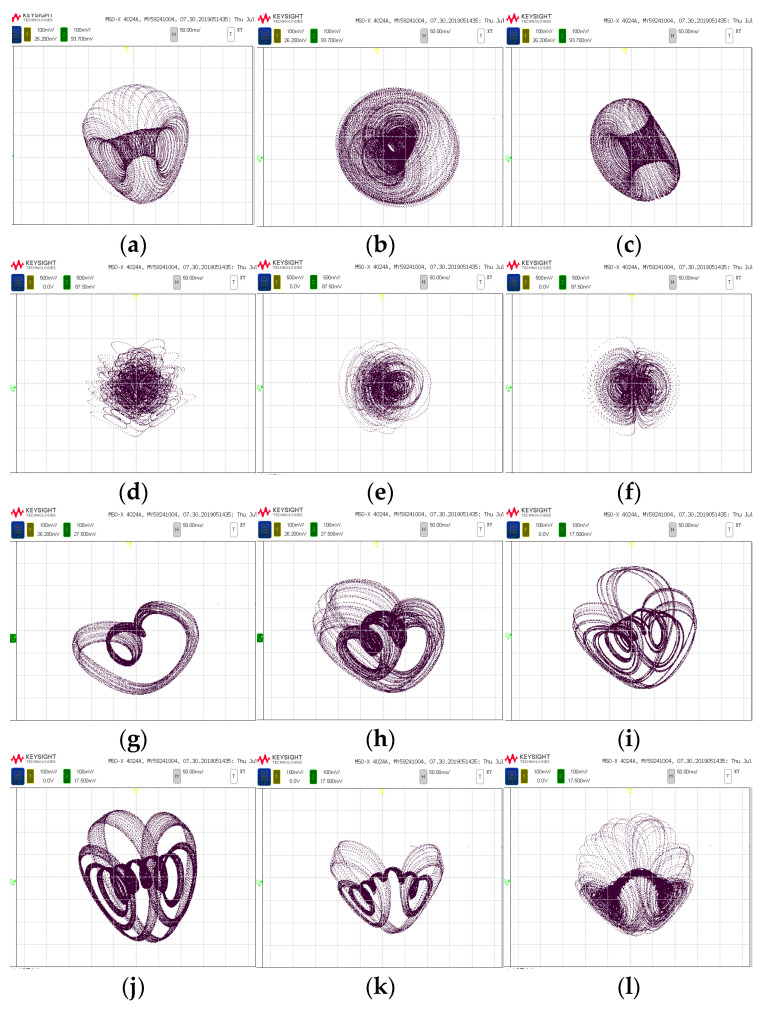
FPGA realization of chaotic system attractor map; (**a**) *x–y* Plane (period); (**b**) *y–z* Plane (period); (**c**) *x–z* Plane (period); (**d**) *x–y* Plane (Chaos); (**e**) *y–z* Plane (Chaos); (**f**) *x–z* Plane (Chaos) (**g**) *b* = 1.2 (**h**) *b* = 1.9; (**i**) *b* = 2.2 (**j**) *c* = 1.5 (**k**) *c* = 3.3 (**l**) *c* = 4.

**Figure 15 entropy-25-01399-f015:**
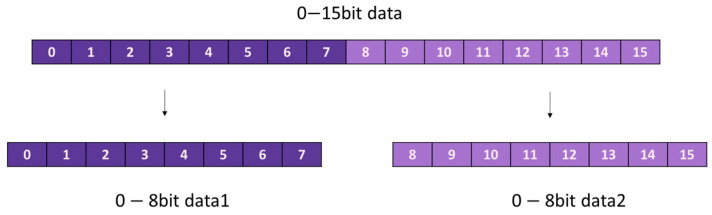
Data decomposition schematic.

**Figure 16 entropy-25-01399-f016:**
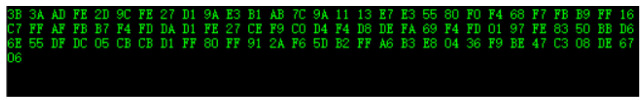
Serial data receiving interface diagram.

**Figure 17 entropy-25-01399-f017:**
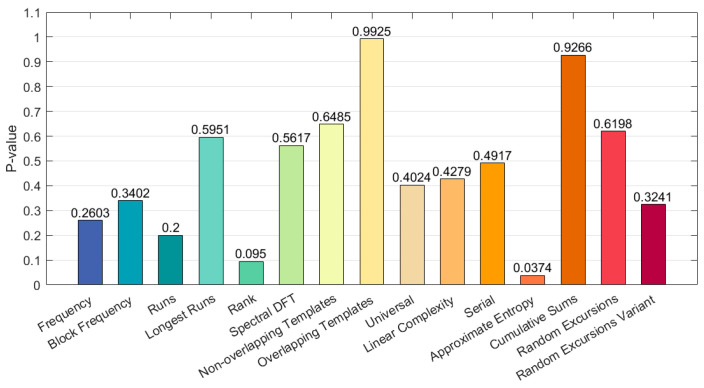
NIST testing of serial data.

**Figure 18 entropy-25-01399-f018:**
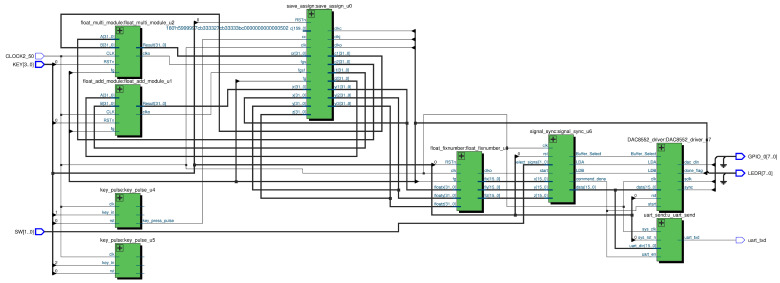
Wiring diagram of the logical structure of the pseudo-random sequence generator.

**Figure 19 entropy-25-01399-f019:**
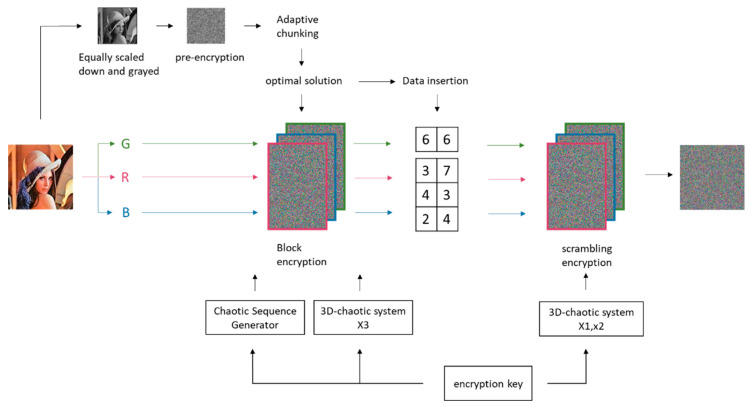
encryption schematic.

**Figure 20 entropy-25-01399-f020:**
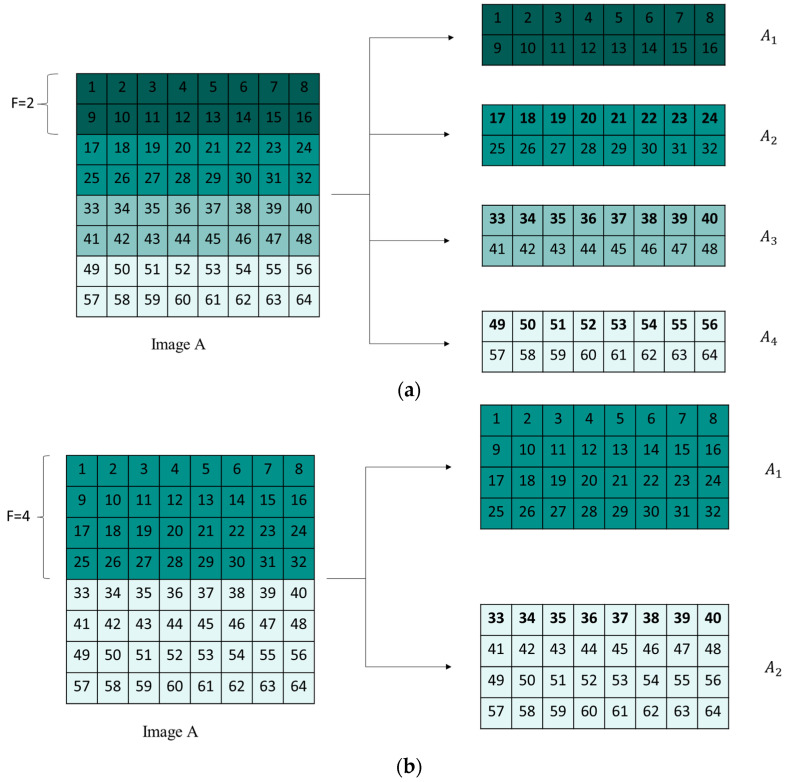
Image chunk process: (**a**) F = 2; (**b**) F = 4.

**Figure 21 entropy-25-01399-f021:**
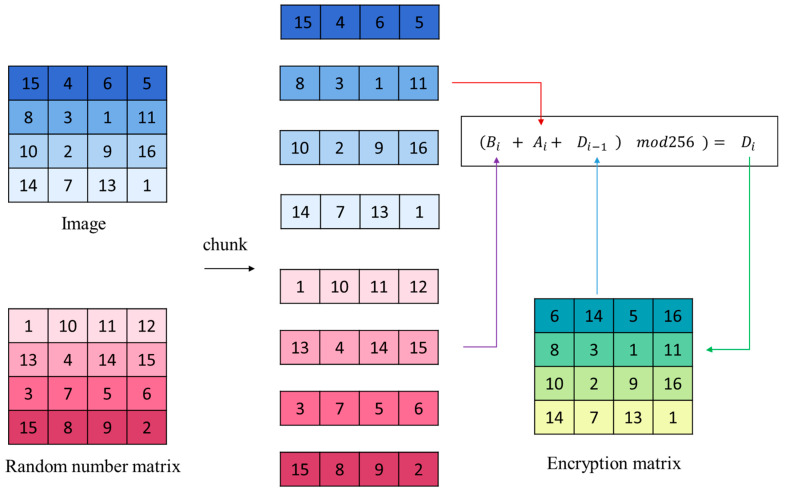
Schematic diagram of chunk encryption.

**Figure 22 entropy-25-01399-f022:**
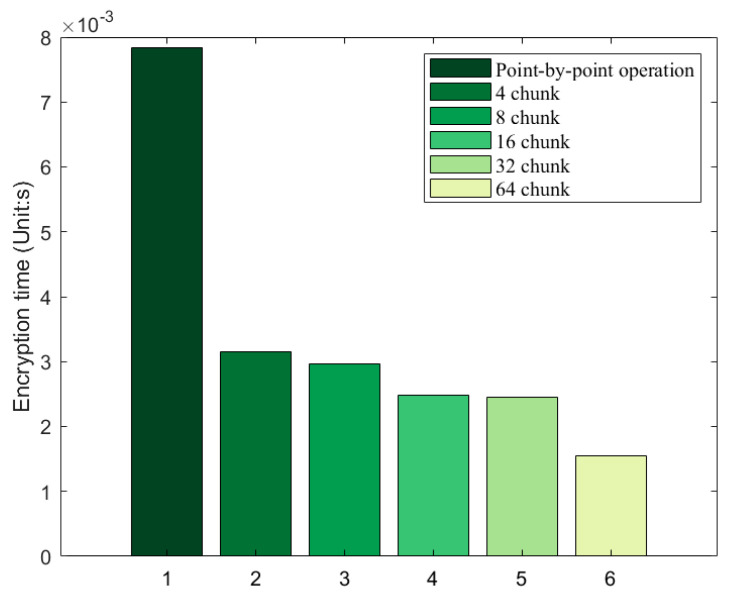
Comparison of encryption times of different chunking policies (Unit: s).

**Figure 23 entropy-25-01399-f023:**
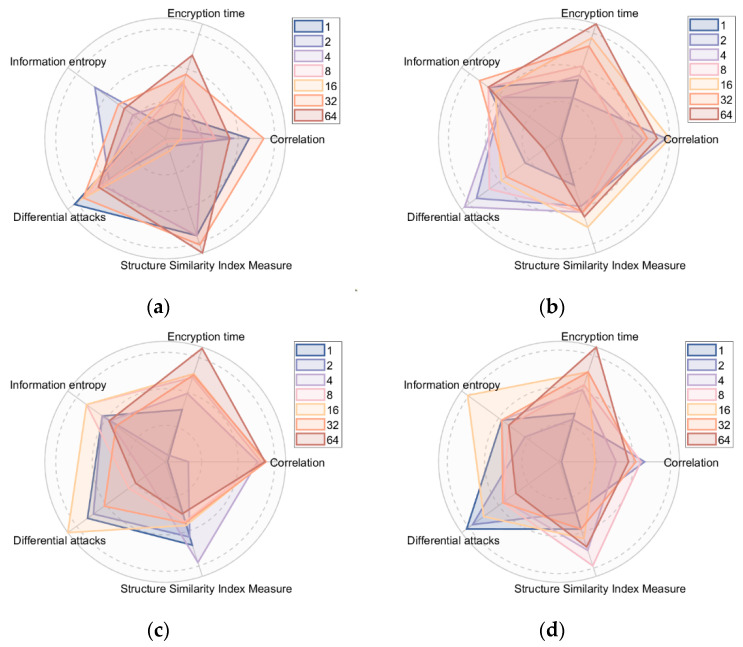
Encryption performance of different images with different chunking strategies: (**a**) Baboon; (**b**) Barbara; (**c**); House (**d**) Pepper.

**Figure 24 entropy-25-01399-f024:**
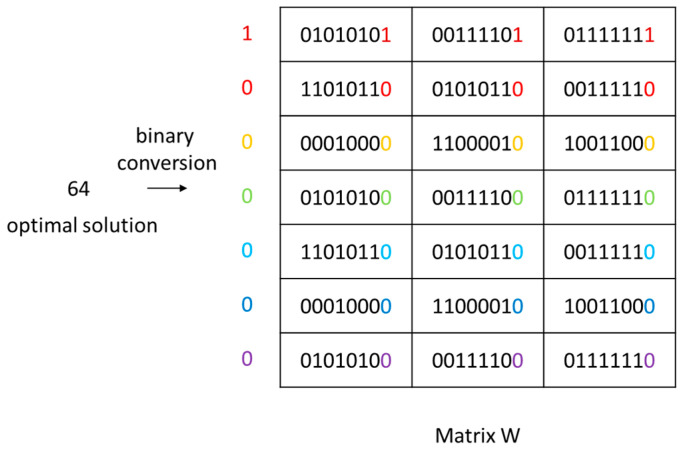
Schematic diagram of data steganography.

**Figure 25 entropy-25-01399-f025:**
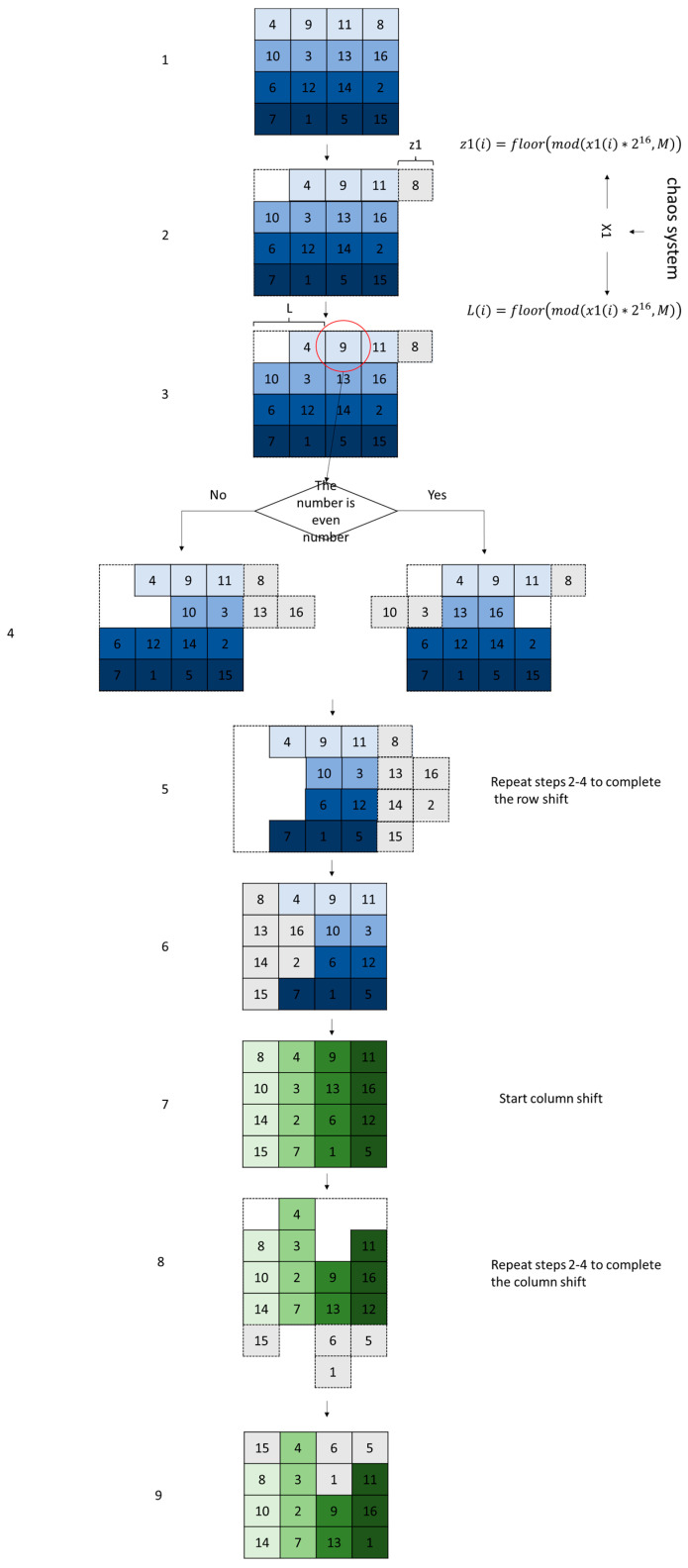
Schematic diagram of disorganization.

**Figure 26 entropy-25-01399-f026:**
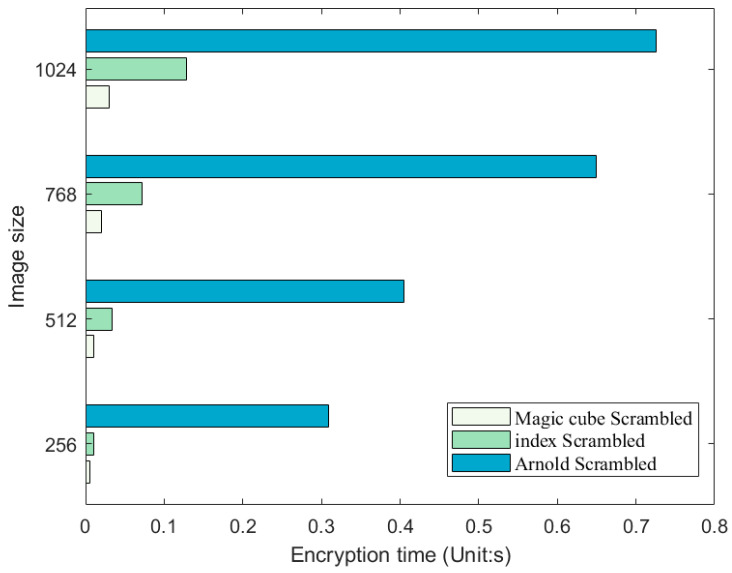
Comparison of encryption times of different scrambling methods (Unit: s).

**Figure 27 entropy-25-01399-f027:**
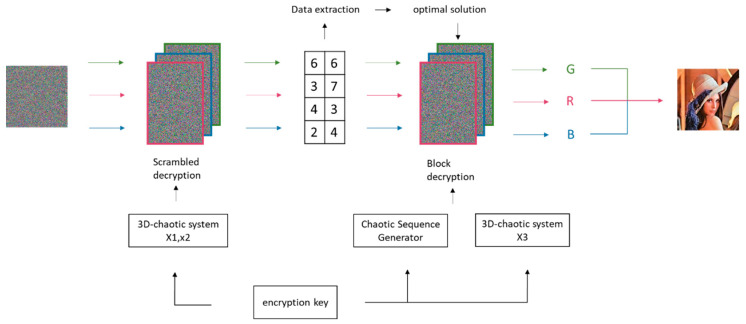
Decryption schematic.

**Figure 28 entropy-25-01399-f028:**
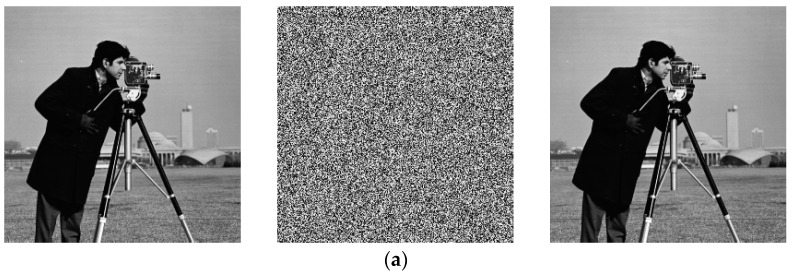
Encryption and decryption renderings of the test images: (**a**) encryption and decryption image of cameraman; (**b**) encryption and decryption image of baboon; (**c**) encryption and decryption image of house; (**d**) encryption and decryption image of airplane (**e**) encryption and decryption image of pepper.

**Figure 29 entropy-25-01399-f029:**
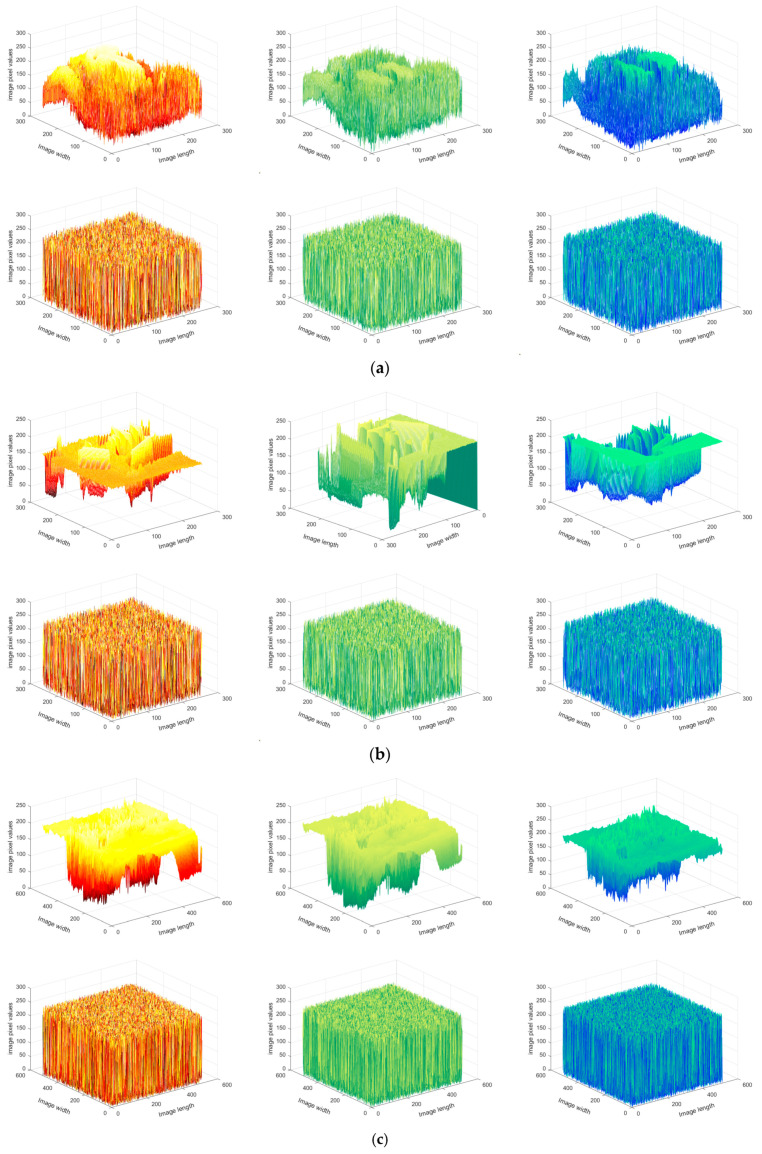
Three-dimensional histograms of the original and encrypted images: (**a**) baboon, (**b**) house, (**c**) airplane, (**d**) pepper.

**Figure 30 entropy-25-01399-f030:**
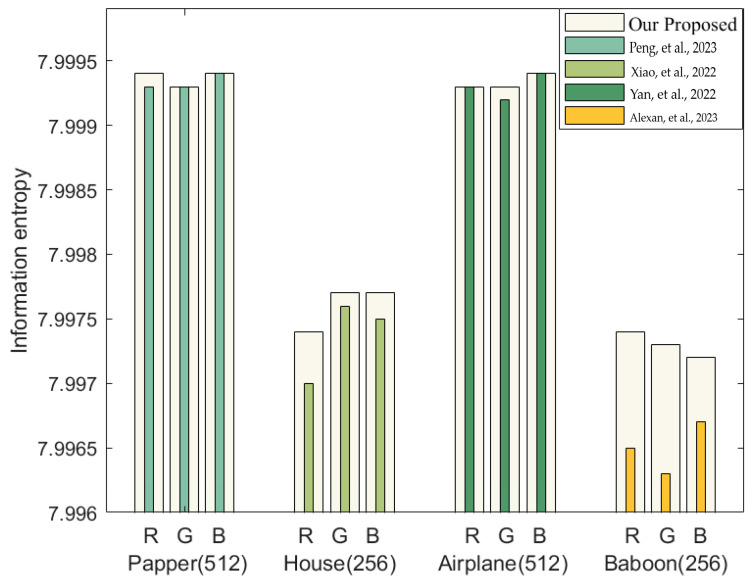
Comparison of information entropy with other algorithms [49,50,51,52].

**Figure 31 entropy-25-01399-f031:**
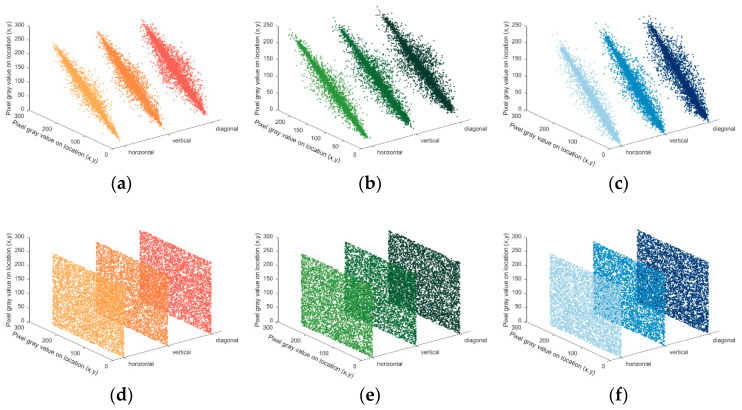
Correlation comparison before and after image encryption. (**a**) R-channel correlation before encryption. (**b**) G-channel correlation before encryption. (**c**) B-channel correlation before encryption. (**d**) R-channel correlation after encryption$E\left(x\right)=\frac{1}{N}\sum_{i=1}^{N}x_{i}$ (**e**) G-channel correlation after encryption. (**f**) B-channel correlation after encryption.

**Figure 32 entropy-25-01399-f032:**
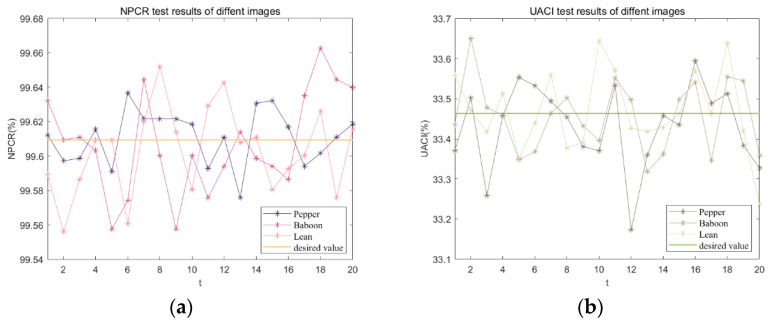
Anti-differential attack test: (**a**) NPCR test results for different images; (**b**) UACI test results for different images.

**Figure 33 entropy-25-01399-f033:**
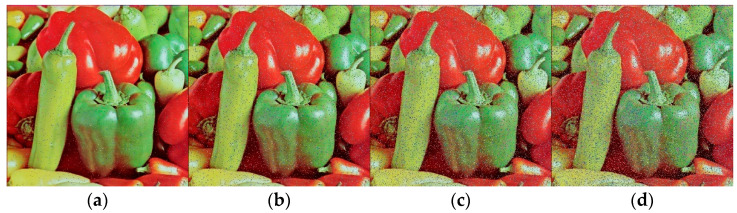
Decrypted image under different levels of noise attack. (**a**) Noise density of 0.01. (**b**) Noise density of 0.05. (**c**) Noise density of 0.1. (**d**) Noise density of 0.15.

**Figure 34 entropy-25-01399-f034:**
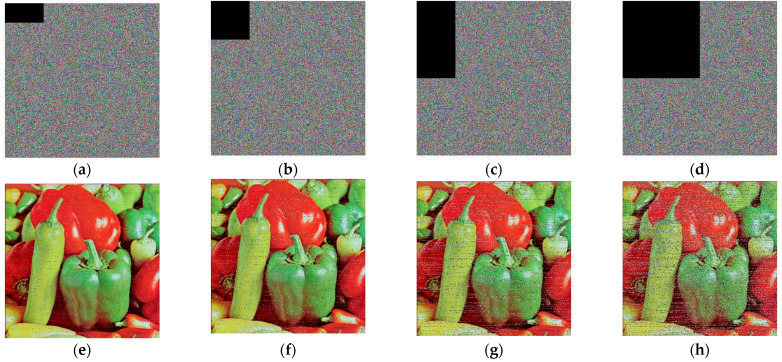
Anti-occlusion attack detection for Pepper image. (**a**–**d**) Various levels of occluded images. (**e**–**h**) Decrypted images of (**a**–**d**).

**Figure 35 entropy-25-01399-f035:**
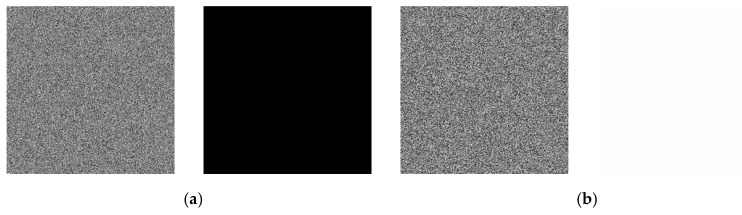
Chosen-plaintext attack test results. (**a**) Encryption and decryption of all-black image. (**b**) Encryption and decryption image of all-white image.

**Figure 36 entropy-25-01399-f036:**
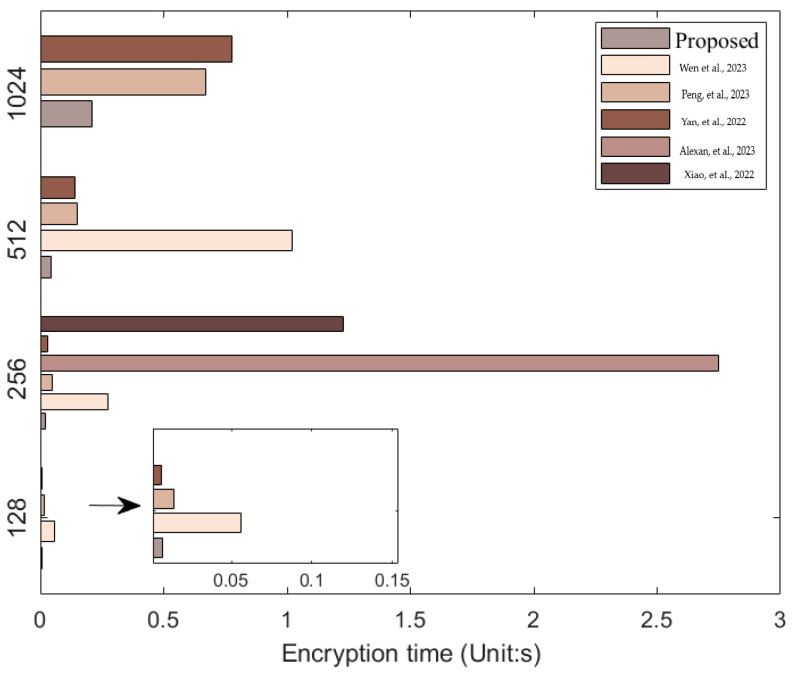
Comparison of encryption times for different algorithms (Unit: s) [49,50,51,52,53].

**Table 1 entropy-25-01399-t001:** NIST test of proposed 3D chaotic system.

Test	*p*-Value (X)	*p*-Value (Y)	*p*-Value (Z)	State
Frequency	0.2653	0.2460	0.9793	PASS
BlockFrequency	0.7785	0.4092	0.4190	PASS
CumulativeSums	0.8331	0.9437	0.6171	PASS
Runs	0.1059	0.9967	0.5615	PASS
LongestRun	0.0240	0.2106	0.6260	PASS
NonOverlappingTemplate	0.9768	0.3383	0.9537	PASS
Serial	0.8222	0.3072	0.1387	PASS
LinearComplexity	0.9772	0.9043	0.9307	PASS
RandomExcursions	0.4870	0.6111	0.8557	PASS
RandomExcursionsVariant	0.5523	0.7880	0.4997	PASS
ApproximateEntropy	0.0785	0.1275	0.6155	PASS
Universal	0.1756	0.5527	0.6179	PASS
FFT	1.0000	0.4009	0.4420	PASS
Rank	0.2272	0.1482	0.4449	PASS
OverlappingTemplate	0.4878	0.2704	0.5078	PASS

**Table 2 entropy-25-01399-t002:** Data decomposition rules.

Input Data	Output Data	
[15-0bit]*x*	[7-0bit]*x*	[15-8bit]*x*
[15-0bit]*y*	[7-0bit]*y*	[15-8bit]*y*
[15-0bit]*z*	[7-0bit]*z*	[15-8bit]*z*

**Table 3 entropy-25-01399-t003:** Evaluation function decision table.

	E1	C1	T1	S1
ideal solution	8	0	0	0
negative ideal solution	6.5	1	10	1

**Table 4 entropy-25-01399-t004:** The data of test images.

Name of the Image	Image Type	Image Size
Cameraman.bmp	Grayscale image	256 × 256
Baboon.jpg	Color image	256 × 256
House.png	Color image	256 × 256
Airplane.bmp	Color image	512 × 512
Pepper.tiff	Color image	512 × 512

**Table 5 entropy-25-01399-t005:** Different images $\chi^{2}$ Test data results.

Image	$\chi^{2}$ Test		
	R	G	B
Baboon (256 × 256)	236.8828	248.0649	255.3281
House (256 × 256)	260.0078	264.8750	223.5625
Airplane (512 × 512)	251.3828	241.2832	252.7246
Pepper (512 × 512)	231.4551	232.3555	248.7051

**Table 6 entropy-25-01399-t006:** Information entropy of plaintext image and ciphertext image.

Image/Size	RGB Components of the Image	Information Entropy of Plaintext Image	Information Entropy of Ciphertext Image
Cameraman.bmp (256 × 256)	-	7.0097	7.9974
	R	7.5856	7.9974
Baboon.jpg	G	7.8284	7.9973
256 × 256	B	7.3319	7.9972
	R	6.4311	7.9974
House.png	G	6.5389	7.9977
256 × 256	B	6.2320	7.9977
	R	6.7178	7.9993
Airplane.bmp	G	6.7990	7.9993
512 × 512	B	6.6390	7.9994
	R	7.3319	7.9994
Pepper.tiff	G	7.5254	7.9993
512 × 512	B	7.0973	7.9994

**Table 7 entropy-25-01399-t007:** Comparison of information entropy with other algorithms.

	Encryption Algorithm	R	G	B
Pepper.png (512 × 512)	Proposed	7.9994	7.9993	7.9994
	Ref. [49]	7.9993	7.9993	7.9994
House.png (256 × 256)	Proposed	7.9974	7.9977	7.9977
	Ref. [50]	7.9970	7.9976	7.9975
Airplane.bmp (512 × 512)	Proposed	7.9993	7.9993	7.9994
	Ref. [51]	7.9993	7.9992	7.9994
Baboon.jpg (256 × 256)	Proposed	7.9974	7.9973	7.9972
	Ref. [52]	7.9965	7.9963	7.9967

**Table 8 entropy-25-01399-t008:** Comparison with correlation coefficients of other studies.

Correlation	This Paper	Ref. [53]	Ref. [49]	Ref. [52]	Ref. [51]	Ref. [50]	Ref. [54]
Red Channel							
Horizontal	−0.0011	0.0030	− 0.0050	0.00073	−0.0049	− 0.0013	−0.0050
Vertical	− 0.0014	−0.0022	− 0.0084	0.00311	−0.0174	− 0.0111	−0.0025
Diagonal	0.0005	0.0006	−0.0062	−0.00508	0.0045	0.0046	0.0035
Green Channel							
Horizontal	−0.0016	−0.0091	0.0163	−0.00054	0.0011	0.0135	−0.0096
Vertical	−0.0004	−0.0129	−0.0101	0.00076	−0.0156	0.0064	−0.0032
Diagonal	0.0007	0.0043	0.0117	0.00331	−0.0160	−0.0241	−0.0023
Blue Channel							
Horizontal	−0.0013	−0.0113	−0.0162	0.00147	−0.0045	0.0179	0.0018
Vertical	−0.0029	−0.0038	0.0273	−0.00147	−0.0175	0.0131	0.0015
Diagonal	−0.0031	−0.0164	0.0256	0.006219	0.0018	0.0023	−0.0042

**Table 9 entropy-25-01399-t009:** Comparison of NPCR and UAIC results for Lena color image.

Encryption Algorithm	NPCR (%)			UACI (%)		
	R	G	B	R	G	B
Proposed	99.6110	99.6068	99.6030	33.4318	33.4552	33.4678
Ref. [49]	99.6826	99.6170	99.5773	33.5152	33.5370	33.3782
Ref. [52]	99.6245	99.6245	99.6245	33.0704	30.7620	27.8720
Ref. [51]	99.6257	99.6145	99.6257	33.4892	33.4798	33.4916
Ref. [50]	99.6429	99.6628	99.6261	33.4440	33.4876	33.4167
Ref. [54]	99.6116	99.6052	99.6070	33.4382	33.4862	33.4426

**Table 10 entropy-25-01399-t010:** Calculation of PSNR and SSIM of images after adding different levels of noise.

Noise Intensity	PSNR	SSIM
0.01	25.4208	0.9454
0.05	19.1361	0.7984
0.1	16.3294	0.6637
0.15	14.7205	0.5627

**Table 11 entropy-25-01399-t011:** Calculation of PSNR and SSIM for images with different levels of Occlusion.

Occluded Degree	PSNR	SSIM
1/32	23.1124	0.9169
1/16	20.1185	0.8483
1/8	17.1374	0.7400
1/4	14.0933	0.5578

**Table 12 entropy-25-01399-t012:** Information entropy and correlation coefficients for test images.

	$\chi^{2}$	Information Entropy	Correlation		
			Horizontal	Vertical	Diagonal
All-black encrypted image	283.5584	7.9993	−0.0049	−0.0051	−0.0033
All- white encrypted image	237.5155	7.9991	−0.0049	−0.0051	−0.0033

**Table 13 entropy-25-01399-t013:** Different-size color image encryption time tests (Unit: s).

Image Types	256 × 256	512 × 512	768 × 768	1024 × 1024
Grayscale image	0.007677	0.017778	0.038957	0.076597
Color image	0.022775	0.046174	0.109098	0.210875

**Table 14 entropy-25-01399-t014:** Comparison of encryption times for different algorithms (Unit: s).

Encryption Algorithm	128 × 128 × 3	256 × 256 × 3	512 × 512 × 3	1024 × 1024 × 3
Proposed	0.006990	0.020775	0.044174	0.208875
Ref. [53]	0.05621	0.27413	1.01921	-
Ref. [49]	0.014233	0.048602	0.151934	0.671352
Ref. [52]	-	2.750966	-	-
Ref. [51]	0.006679	0.03156	0.142001	0.775764
Ref. [50]	-	1.2271	-	-

## Data Availability

Not applicable.

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
