# Peer review of "Adaptive Fast Image Encryption Algorithm Based on Three-Dimensional Chaotic System"

_entropy, 2023, doi:10.3390/e25101399_

Round 1

Reviewer 1 Report

In this paper, a new three-dimensional chaotic system is proposed, and it can be seen that the system shows diverse dynamic behaviors as the parameters change, which is an interesting study. At the same time, an image encryption algorithm based on the chaotic system model is designed, and some application potential of the algorithm in image security is shown through tests. According to the original draft, I have the following suggestions for improvement.

1. I have problems with the following parts of the article, such as:

1) Figure 9 Table title (b) is misnamed and should be C0(x0∈[-3, 3], y0∈[-3, 3])

2) The variables in Figure 2-7 should be in italics

3) Figure 10 Table titles are incorrectly named

4) In 2.8 summaries the key sensitivity parameter for this variable is 10-15 unreasonable

5) i=else in formula 7 is not a good description

6) The description of j∈[1] in formula 17, 18 is not appropriate, it can be changed to j=1

7) The expression of variable i in formula 19 DZ(i)=cirleftshift(PBK(i),z1(i)) i∈[1,M] is unreasonable and in conflict with the expression of the following formula

8) The unit of encryption time in Figure 24 needs to be indicated in the figure title

9) What test criteria were not specified in 2.9.2 Random test of serial port data in small pairs.

10) 4.8 Summary The title name is different from the description below. This part is "Occlusion attack" or "cropped attack"

2. In the conclusion section, we can briefly look forward to the application prospects of the algorithm, such as whether the algorithm can be used in real-time video encryption and other fields.

In short, the overall content of this paper is relatively complete and the workload is rich, but there are still some details that can be further improved. I have given some specific modification suggestions, hoping to provide some references. Looking forward to the effect of the revised paper!

Author Response

Dear  Reviewers,

Since there are a lot of formulas and images in the text box that won't display, I wrote my response in a word document. Thank you very much for taking the time to review this manuscript.

Reviewer 2 Report

The article concerns a new chaotic system, its FPGA implementation, and its use in image encryption.

The article is interesting and topical. However, authors should pay attention to the style of presentation of algorithms. They are often described in little detail, making them difficult to understand.

Some detailed comments:

1) Citations in the text are in the indexes.

2) The text in some places is written in a strange style, e.g., lines 45-52.

3) I consider the literature review to be insufficient, both regarding dynamic systems and encryption algorithms.

4) Line 59: Move the sentence "The remainder of this..." to a new paragraph

5) When determining LE, it would be useful to have formulas that show how to obtain this measure. The same note applies to the measures in section 2.6.

6) Some of the charts are of very poor quality, e.g. Figure 2a,b. For bifurcation diagrams, I suggest choosing a denser grid of parameter values for its calculation.

7) 2.7 uses NIST tests to check exactly what?

8) In formulas, I suggest using another sign for multiplication instead of *. Additionally, a comment regarding the improved Euler algorithm would be appreciated or at least a reference to the literature. I also suggest using variables other than H_{i,j}.

9) Are not some of the charts in Figure 14 already presented earlier, e.g., in Figure 1?

10) I don't understand what Table 2 shows. Please explain.

11) Figure 18: What exactly is tested with NIST tests? How long is this binary string? Is it just one string?

12) How are steps 3 and 4 of the encryption algorithm related?

13) What exactly does the division into blocks from step 5 look like?

14) I propose to write formula (7) using standard algorithmic and mathematical expressions. The same remark applies to further formulas.

15) Some charts do not have axes labeled, e.g. Figure 27.

16) Formula (22) - the "-" sign is missing.

17) Formula (23) - wrong denominator.

Author Response

(The authors gave the same response as above.)

Reviewer 3 Report

In the peer-reviewed manuscript, the authors present a novel three-dimensional chaotic system that demonstrates sensitivity to parameter variations, exhibiting diverse phase trajectories, offset behaviors, and expansion-contraction phenomena. Their model covers a broad chaotic range and proves suitable for integration within image encryption. In terms of encryption algorithm design, this paper presents a fast image encryption algorithm with an adaptive mechanism.

Experimental results underscore the superior performance of this algorithm in terms of both encryption duration and security.

The paper brings original novel information in the domain of the journal’s thematic focus. The research results are clearly distinguished from results adopted and used literary resources are mentioned properly. Credibility of published results is documented (experiments - simulations). Text readability and its linguistic correctness (even English texts, especially in the case of the technical terminology) is on the appropriate level.

I do not have any comments on the scientific content of the article!!!

I have comments on the formal side of the article:

1. keywords need to be arranged alphabetically,

2. the authors do not refer to every declared literature in the text, e.g. Reference no. 5, 6, etc.,

3. the format of references to literature is not uniform,

4. not all images are legible, e.g. Fig. no. 12, 15, 16, etc..

In general, after appropriate corrections and additions, I approve publication of this manuscript.

Author Response

(The authors gave the same response as above.)

Round 2

Reviewer 2 Report

Thanks to the authors for their comprehensive answers. In my opinion, the article has improved in quality. I have the following additional comments:

1) I propose to present the description of the algorithms more concisely. It is better to move additional comments beyond the description of the algorithm itself, e.g. points 4 and 5 from algorithm 3.1.

2) the description of step 5 in the above algorithm does not say at all what should be done at this stage as part of the encryption process.

3) I suggest expanding the number of keywords so that the article can reach a larger audience.

Author Response

(The authors gave the same response as above.)
